# Migratory Take-Off Behaviour of the Mongolian Grasshopper *Oedaleus asiaticus*

**DOI:** 10.3390/insects11070416

**Published:** 2020-07-04

**Authors:** Yun-Ping Wang, Xiong-Bing Tu, Pei-Jiong Lin, Shuang Li, Chao-Min Xu, Xin-Qiao Wang, Don R. Reynolds, Jason Chapman, Ze-Hua Zhang, Gao Hu

**Affiliations:** 1College of Plant Protection, Nanjing Agricultural University, Nanjing 210095, China; 2017102066@njau.edu.cn (Y.-P.W.); 2018102071@njau.edu.cn (P.-J.L.); J.Chapman2@exeter.ac.uk (J.C.); 2State Key Laboratory for Biology of Plant Diseases and Insect Pests, Institute of Plant Protection, Chinese Academy of Agricultural Sciences, Beijing 100193, China; xbtu@ippcaas.cn (X.-B.T.); sclishuang61@163.com (S.L.); 18363972673@163.com (C.-M.X.); 3Lushan Meteorological Bureau, Jiujiang 332900, China; w18807027897@163.com; 4Natural Resources Institute, University of Greenwich, Chatham, Kent ME4 4TB, UK; D.Reynolds@greenwich.ac.uk; 5Rothamsted Research, Harpenden, Hertfordshire AL5 2JQ, UK; 6Centre for Ecology and Conservation, and Environment and Sustainability Institute, University of Exeter, Penryn, Cornwall TR10 9FE, UK

**Keywords:** *Oedaleus asiaticus*, take off, insect migration, weather conditions, inner Mongolia

## Abstract

*Oedaleus asiaticus* is one of the dominant species of grasshoppers in the rangeland on the Mongolian plateau, and a serious pest, but its migratory behavior is poorly known. We investigated the take-off behavior of migratory *O. asiaticus* in field cages in the inner Mongolia region of northern China. The species shows a degree of density-dependent phase polyphenism, with high-density swarming populations characterized by a brown morph, while low-density populations are more likely to comprise a green morph. We found that only 12.4% of brown morphs engaged in migratory take-off, and 2.0% of green morphs. Migratory grasshoppers took off at dusk, especially in the half hour after sunset (20:00–20:30 h). Most emigrating individuals did not have any food in their digestive tract, and the females were mated but with immature ovaries. In contrast, non-emigrating individuals rarely had empty digestive tracts, and most females were mated and sexually mature. Therefore, it seems clear that individuals prepare for migration in the afternoon by eliminating food residue from the body, and migration is largely restricted to sexually immature stages (at least in females). Furthermore, it was found that weather conditions (particularly temperature and wind speed at 15:00 h) in the afternoon had a significant effect on take-off that evening, with *O. asiaticus* preferring to take off in warm, dry and calm weather. The findings of this study will contribute to a reliable basis for forecasting migratory movements of this pest.

## 1. Introduction

Since the beginnings of civilization, grasshoppers, including locusts (Orthoptera: Acridoidea), have been among the most devastating agricultural pests [1,2,3,4]. This group of insects contains hundreds of pest species and affects the livelihoods of one in every ten people worldwide [2]. Some species can migrate hundreds of kilometers per day and destroy all green vegetation over millions of square kilometers in a very short time [1,2,3,5]. Therefore, an understanding of their migration behaviour is crucially important for the forecasting and control of many acridoid pests and, in fact, this behavior is well known in some species, such as the desert locust (*Schistocerca gregaria*) [6,7] and oriental migratory locust (*Locusta migratoria manilensis*) [8]. The Mongolian grasshopper (*Oedaleus asiaticus* Bei-Bienko; sometimes referred to as *Oedaleus decorus asiaticus* [9]) is one of the most dominant species in grassland and pastoral areas in Mongolia and northern China, where there have been serious outbreaks in recent decades [3,10,11,12]. Despite this, its migratory behavior has been scarcely studied and is relatively poorly known.

*Oedaleus asiaticus* is mainly distributed in semi-desert and grassland steppes on the Mongolian Plateau; it is highly abundant, often accounting for 50% of the local grasshopper community, and sometimes up to 90% [12,13]. It not only feeds on many pasture species in the Gramineae, Iridaceae and Cyperaceae families, but also on various crops, such as corn, sorghum, wheat, and millet crops [13,14]. With the gradual exacerbation of climate change and heavy livestock pressure, the grasshopper problem has become more serious, resulting in major threats to agricultural and animal husbandry production in northern China, especially in inner Mongolia [10,15]. 

*Oedaleus asiaticus* adults have a strong flight capability, flying for more than 2 h with an accumulated flight distance of up to 15 km in laboratory conditions [16], which would likely translate into hundreds of kilometers within a few days under natural conditions [17], particularly if they engage in high-altitude windborne movements. *Oedaleus asiaticus* shows a degree of density-dependent phase polyphenism and has two morphs: the darker, brown-coloured morph is more characteristic of high-density outbreaking populations and the green morph of non-outbreak populations [18,19]. Individuals of the brown morph have increased relative investment in their thorax and hind-legs, and mostly appear during outbreak conditions, although the link between the colour morphs and migratory propensity is not always clear-cut [18]. Several migration events of *O. asiaticus* have been observed in previous studies [15,17]. For example, many *O. asiaticus* adults were observed to suddenly appear in many sites from north to south successively in southern inner Mongolia (Chifeng urban area and Duolun county), northern Hebei (Zhangjiakou and Chengde urban areas) and Beijing during early and mid-July 2002, and also to disappear abruptly about seven days later at all these sites [17]. Most of the sites (e.g., northern Hebei and Beijing) do not have suitable habitats for *O. asiaticus* to survive year-round [9,10], and no nymphs were found before the adults appeared, thus these grasshoppers must have migrated from elsewhere [17]. Moreover, *O. asiaticus* was observed to take off in the evening and land in the early morning [17], indicating it makes nocturnal migrations over several nights. It is well known that many migratory grasshopper species (and solitarious locusts) commence migratory flight at dusk, and fly for a variable period during the night [20,21,22]. Nevertheless, compared with other destructive grasshoppers and locusts, there is little knowledge of the migration behavior of *O. asiaticus*, which is hampering management of this serious migratory pest. 

In this study, we investigated the take-off behavior of migratory *O. asiaticus* in a field cage where we could explore the relationship between take-off and weather conditions. The aim was to get some basic knowledge of the migration biology (e.g., diel timing of emigration, proportion of the population leaving, physiological state of the migrants) of *O. asiaticus* and identify the key meteorological factors that promote emigration. 

## 2. Materials and Methods

### 2.1. Overview of the Field Site

Our experiment was carried out in the Scientific Observing and Experimental Station of Pests in Xilingol Rangeland, Ministry of Agriculture, P. R. China, located in the western suburb of Xilinhot City, inner Mongolia province (43°57′ N, 116°00′ E). The test grasshoppers were collected from the grassland of Hanniwulagacha near the experimental station. The grassland soil is chestnut soil, and the vegetation is mainly the grass *Leymus chinensis*. The experimental area is a typical steppe with a semi-arid continental monsoon climate in the middle temperate zone. The annual precipitation is 267.9 mm, the annual average temperature is 3.1 °C, and the altitude is 1024 m above sea level. At the beginning of June each year, *O. asiaticus* begins to hatch from egg pods, goes through five instars, and then moults into the adult by mid-July. The adult lifespan is about 4–9 weeks [13,15].

### 2.2. Take-Off Behaviour Observations

Before the start of the experiment, a 2 m × 2 m × 2 m field cage was built in an outdoor open space. Twenty plants of *Stipa krylovii* Roshev were moved to the field cage to feed the grasshoppers. *Oedaleus asiaticus* nymphs and adults were caught every few days and kept in small indoor net cages (a cylinder cage with diameter 24 cm and height 26 cm) with *S. krylovii* plants as food. To keep the small net cages close to the natural state, no temperature and humidity measuring instruments were used indoors. 

Our experiment was carried out during 14 July–6 August, in total 24 days. On 14 July, *O. asiaticus* adults, comprising 100 brown phenotype (1♀∶1♂) and 20 green (1♀∶1♂), were moved from the small net cages into the field cage in the morning (about 07:00–08:00 h local time) providing a similar population density and % phenotype as under natural conditions (i.e., about 30 individuals per m^2^, comprised of 80% of the brown phenotype). One observer entered the field cage at 18:30 h every evening to watch the grasshopper take-off. Individuals that took off several times and climbed up to a height of >1 m were identified as “take-off individuals”. These take-off individuals were caught with a tube and removed from the field cage, and their number was counted every half hour. The observations were ended at 21:00, and then a sample of 20 non-take-off brown grasshoppers (1♀∶1♂) and 10 non-take-off green grasshoppers (1♀∶1♂) were caught and removed from the field cage. New grasshopper adults were placed in the field cage the next morning to keep the number of test grasshoppers the same. All the females and males removed from the field cage were dissected after observation each day. The presence of residual food in their digestive system, the degree of ovarian development, the mating status, and any parasitization were determined. The degree of ovarian development was classified into five levels according to Han Haibin’s method [23]. Females with ovaries at Level I or II are still immature and not ready for oviposition; the ovaries of matured females are at Level-III or above. The mating status of females was estimated by checking for the presence or absence of a spermatophore. Individuals were identified as “With Food” if any residual food was found in the digestive tract, or identified as “No Food” if the digestive tract was transparent (Figure 1E).

### 2.3. Regression Models with Meteorological Data

Hourly surface meteorological data in July and August 2019 were obtained from a meteorological station in Xilinhot city (116.064° E, 43.953° N, about 5 km away from our experimental site). To explore the relationship between grasshoppers’ take-off behaviour and weather conditions, we calculated a series of Spearman’s correlation coefficients between the daily number of take-off grasshoppers with several meteorological factors in the pre-take-off (13:00–18:00 h) and the take-off periods (19:00–21:00 h). These meteorological factors included air pressure, the three-hour pressure change, wind direction, wind speed, air temperature, total precipitation within 1-, 3-, 6- and 12-h, and relative humidity. All variables except precipitation are the mean value in one hour. Wind direction was incorporated as the cosine (cos) and sine (sin) of the wind direction, to convert to a linear measure representing the degree of “northerliness/southerliness” and “westerliness/easterliness” of the wind respectively, where 1 = blowing from the north and −1 = from the south in cosine of wind direction, and 1 = blowing from the east and −1 = from the west in sine of wind direction. Therefore, in total, 99 (11 variables × 9 h) correlation coefficients were calculated (Figure 2). 

Based on the above correlation coefficients, the variables significantly correlated with the daily number of take-off grasshoppers were chosen as the potential explanatory variables to build a regression model. Data exploration was applied following the protocol described by Zuur et al., (2010) [24]. The presence of outliers, auto-correlation in the response variables and collinearity were examined. The variance inflation factor (VIF) was used to measure the multicollinearity, and a potential variable was excluded if its VIF value was >5 [24,25]. As the initial analysis indicated that the residuals and the response variables were not auto-correlated, a Poisson generalized linear model (GLM) was applied. Due to the small sample size, a forward selection was applied using the Akaike information criterion (AIC). Auto-correlation and over-dispersion in the Pearson residuals of the fitted model were checked, justifying the use of a Poisson GLM. 

### 2.4. Statistical Analysis

Chi-square tests were applied to test the significance of the take-off percentage between females and males, brown and green morphs, and to determine the significance of the parasitism rate, status of digestive tract and sexual maturity between take-off and non-take-off individuals. When sample sizes were less than 5, Fisher’s exact test was applied. The distribution of wind direction was tested by Rayleigh test, and mean direction and r - value were calculated. The Rayleigh r-value ranges from 0 to 1, with higher values indicating a greater clustering of directions around the mean. As the daily number of take-off *O. asiaticus* did not coincide with a normal distribution, Spearman’s correlation coefficients were calculated between daily take-off numbers and meteorological factors. All chi-square tests, Fisher’s exact test, Rayleigh test, correlation analyses and regression model were applied in R (version 3.6, https://www.r-project.org/).

## 3. Results

### 3.1. Observation of the Take-Off of O. asiaticus

From 14 July to 6 August 2019, 660 adult grasshoppers were used in the experiments, of which 200 were the green-coloured morph. In total, 61 individuals (9.24%) were observed to take off at dusk; 57 of these were brown (12.39% of the 400 brown morphs) and only four of these were green (2.00% of the 200 green morphs). This result indicates that green-coloured individuals have a very weak migration propensity, and are significantly different from the brown morph in their take-off propensity (chi-squared test: *X*^2^ = 17, df = 1, *p* < 0.001) (Figure 1A). Therefore, the green morph of *O. asiaticus* was regarded as resident and it was excluded from the rest of the analyses (although we cannot rule out the possibility that its take-off conditions were merely different from the brown morph). 

Among the 57 brown grasshoppers which took off at dusk, slightly more females than males took-off, but this difference was not quite significant (female: 14.78% (34/230), male: 10.00% (23/230); chi-squared test: *X*^2^ = 3, df = 1, *p* = 0.113) (Figure 1A). 

The take-off observations were carried out from 18:30 to 21:00 h, while the sunset in Xilinhot was at 20:04 h on 14 July, at 19:43 h on 6 August. We found that 75.43% (43/57) individuals took off during the period of 20:00–20:30 h (Figure 1B), and this was significantly higher than at other periods (chi-squared test: *X*^2^ = 116, df = 1, *p* < 0.001). 

During the daytime, grasshoppers were observed to be very active, and can easily be simulated to take off by any disturbance, such as humans approaching the cage or other individuals flying. However, in this “trivial” flight activity, individuals make horizontal flights at a low height (<1 m) and land in a short time, so this behaviour was clearly different from the migratory take-off at dusk.

### 3.2. Take-Off Related to the Physiological State and Parasitization Rate of O. asiaticus

All 57 take-off individuals and 380 non-take-off individuals (190 females and 190 males) were dissected. Of the take-off individuals, 91.22% (52/57) did not have any food remaining in their digestive tract, while only nine non-take-off individuals (2.37%) had an empty digestive tract (chi-squared test: *X*^2^ = 319, df = 1, *p* < 0.001) (Figure 1C,E). 

Among all the 437 dissected individuals, 14 non-take-off individuals were parasitized by sarcophagid flies, but none of take-off individuals (Figure 1C,E). Nonetheless, the results of Fisher’s Exact Test (*p* = 0.232) did not show that parasitization was significantly associated with migration propensity. 

Among 34 dissected take-off females, 70.59% (24/34) were immature, i.e., their ovarian development levels were below Level-III (Figure 1D). This was quite different from non-take-off females (chi-squared test: *X*^2^ = 22, df = 1, *p* < 0.001), where 138 out of 190 females (72.63%) were mature (i.e., their ovarian development levels were at Level III and above) (Figure 1D). All take-off females were found to be mated, and 96.31% (183/190) of non-take-off females were also mated (Figure 1D). The mating rate was thus not different between take-off and non-take-off individuals (Fisher’s Exact Test: *p* = 0.598). 

### 3.3. Take-Off Behaviour of O. asiaticus and Weather Conditions 

During the 14 July to 6 August 2019 period, the weather in Xilinhot was quite dry, the daily relative humidity was only 49.1% (range 30.1%–64.8%). The mean daily temperature was 24.7 ± 0.4 °C (mean ± stand error (S.E.), n = 24 days), but the difference between day and night was large (13.2 ± 0.5 °C, n = 24 days). Thus, the daily max temperature was up to 31.2 ± 0.5 °C (n = 24 days). Surface wind blew towards the south (Rayleigh test; direction 187°, speed 3.3 ± 0.2 m/s, r = 0.59, *p* = 0.0001, n = 24 days).

During the take-off time period (19:00–21:00 h), only air temperature affected the take-off behaviour of the grasshoppers (Figure 2); the daily number of individuals taking off increased significantly with the mean temperature (ρ = 0.640, *p* = 0.0008, df = 23; mean temperature: 25.7 ± 0.6 °C (range 20.0–29.7 °C, n = 24 days)). Because take-off individuals must have had time to empty their digestive tracts, the decision to emigrate must be made earlier in the day and, in fact, we found that the weather in the afternoon (14:00–18:00 h) had a significant effect on the daily number of individuals taking off in the evening (Figure 2). Warmer air temperatures in the 15:00–18:00 h period (mean: 29.3 °C, range: 23.4–33.4 °C, n = 24 days) promoted grasshopper take off, while higher relative humidity in the 14:00–18:00 h period (mean: 32.1%, range: 13.8%–60.4%, n = 24 days), wind speed in 15:00–16:00 h (mean: 4.6 m/s, range: 1.6–8.3 m/s, n = 24 days) and pressure changes during three periods (14:00–16:00 h, 15:00–17:00 h and 16:00–18:00 h) inhibited take-off. An absence of pressure change means the weather will remain stable, and taken together, indicates that the grasshoppers prefer to take off in warm, dry and calm weather. 

Rain in Xilinhot is very rare in summer; out of the 216 h (9 h × 24 days) of our study periods, the precipitation was zero in 206 h (95.37%). The only significant precipitation variable was the total precipitation in last 12 h at 17:00 (i.e., the sum of hourly precipitation during 5:00–16:00 h), which had a negative effect on take-off (Figure 2), probably because of the increased humidity arising from the earlier rain. We did not find that wind directions and air pressure variables affected the take-off behaviour of the grasshoppers significantly (Figure 2). 

The optimized GLM model indicated that wind speed and air temperature at 15:00 h were the most important variables at predicting the daily number of grasshoppers taking off (Table 1). Grasshopper take-off in the evening increased with increasing air temperature at 15:00 h, and declined with increasing wind speed at 15:00 h (Table 1), and this result was consistent with the correlation analyses. In addition, the GLM model also showed that the total precipitation in last 12 h at 17:00 h had a negative effect, but marginally (Table 1). The “generalized R^2^” for this optimized model = 0.62, and AIC = 87.

## 4. Discussion

In this study, the take-off of migratory *O. asiaticus* was investigated in a field cage. Just like most other nocturnal migratory insects with a similar or larger body size, *O. asiaticus* also take off at dusk, especially in the half hour after sunset [20,26,27]. More specifically, the emigration timing was very similar to other migratory grasshoppers such as *O. sengalensis* and *Aiolopus simulatrix* in the African Sahel, where take-off started about 20 min after sunset and peaked 15–30 min later [20,21,22]. In *O. asiaticus*, the take-off percentages were similar in both sexes, and the ovaries of emigrating females had not matured. Again, this is similar to *O. senegalensis*, where long-range migration was most likely to occur in young individuals after the cuticle had hardened but before the females started to mature their first egg batch [20,28]. However, there are some interesting differences between *O. asiaticus* and similar migrants that warrant further discussion: (i) the take-off percentage of *O. asiaticus* was relatively low, (ii) most take-off individuals emptied their digestive tract, and (iii) most take-off females were mated but not sexually mature.

The percentage of *O. asiaticus* that initiated migratory flight was low. Only 12.4% of the brown morph, and 2.0% of the green morph were observed to take off; and furthermore, a previous study [18] found that neither morph would fly in laboratory and field trials. This is different from *O. senegalensis* in the West African Sahel where, in each of the three generations, the whole population emigrates and can move rapidly, sometimes as much as ~350 km in a night [7,20,22].

It is well known that locusts such as *L. migratoria* and *S. gregaria* represent an impressive example of migratory polyphenism, with distinct solitarious and gregarious phases [1,8,29,30]. The gregarious phase exhibits behavioural, morphological and physiological traits associated with day-flying swarm activity, while the solitarious phase is capable of long-range windborne migration at night [1,8,21,29]. The significantly higher take-off percentage of the brown morph of *O. asiaticus* indicates that this form has the greater migration propensity, and this is supported by the observation that all catches of *O. asiaticus* by searchlight trap in northern Beijing province were brown (Zhi Zhang, unpublished data). The fairly low take-off percentage may indicate that the brown morph grasshoppers are not necessarily morphologically and physiologically specialized for migration as discussed in a previous study [18]. The study by Cease et al. found that the brown and green forms were not different in terms of flight muscle mass or lipid store [18], although high population densities could cause both brown and green forms to exhibit some morphological traits associated with migration, such as a larger hind wing area. Genetic differentiation of *O. asiaticus* between populations in inner Mongolia shows a positive correlation with geographical distance, and this also indicated that long-distance migration is not very common in *O. asiaticus* [31]. In our study, the population density was about 30 individuals per m^2^, but densities can be as high as ≥500 individuals per m^2^ in outbreak sites [15,16]. Individuals from higher density populations have stronger flight capability [16], and this implies that take-off percentage would have been greater if our experimental population had been more crowded.

Most emigrating individuals did not have any food remnants in their digestive tracts, even under conditions with sufficient food. It seems that migratory individuals stop eating and eliminate food residue from their body before commencing migration presumably to reduce the body weight and save energy for long-distance flight. This, in turn, means they decide to emigrate some time before the dusk take-off, and this supposition is supported by the finding that the weather conditions in the afternoon (14:00–18:00 h) have a significant effect on the daily number of grasshoppers taking off. There is evidence in other migratory acridoids (e.g., *S. gregaria* and *Chortoicetes terminifera*) that take-off frequency is inversely associated with the quantity of food in the gut [32,33].

Generally, insect migration flight is initiated during the pre-reproduction stage when ovarian development and mating behaviour are suppressed [34,35,36,37]. Therefore, migrating individuals are usually sexually immature and not mated [28,35,37]. However, most take-off females in our study of *O. asiaticus* were mated but not yet sexually mature, which is not consistent with most other migratory insects (e.g., *Mythimna separata* [38] and *Nilaparvata lugens* [39]) that have been studied in detail. Female *O. asiaticus* mating while sexually immature may be a strategy which allows for migration to occur before the development of gravid ovaries (which would significantly increase weight, drag and fuel use during migratory flight), but ensuring that any immigrant females arriving in a new environment are still able to reproduce.

In this study, the relationship between take-off behaviour and weather conditions was also explored. The weather in the afternoon preceding take-off at dusk is important, as migratory *O. asiaticus* need time to empty their digestive tracts. We found that the air temperature, relative humidity, wind speed and three-hour pressure change were significantly correlated with the daily numbers of grasshoppers taking off. Here, little change in the three-hour pressure change indicates stable weather, and shows that grasshoppers prefer to migrate in warm, dry and calm weather, like many other migratory insects [40,41,42]. The weather conditions in the afternoon identified as important for *O. asiaticus* take off will provide information for forecasting the migration of this pest.

## 5. Conclusions

Our study investigated the take-off behaviour of emigrating *O. asiaticus*. The grasshoppers always take off at dusk, within about half an hour after sunset, but only a tenth of individuals of the more migratory brown morph initiate migratory take off. Emigrating individuals prepare for take-off in the afternoon by eliminating food residue in their digestive tracts, and female migration is restricted to mated and sexually immature individuals. Weather conditions in the afternoon, especially temperature and wind speed, significantly affected the number of individuals taking off in the evening.

## Figures and Tables

**Figure 1 insects-11-00416-f001:**
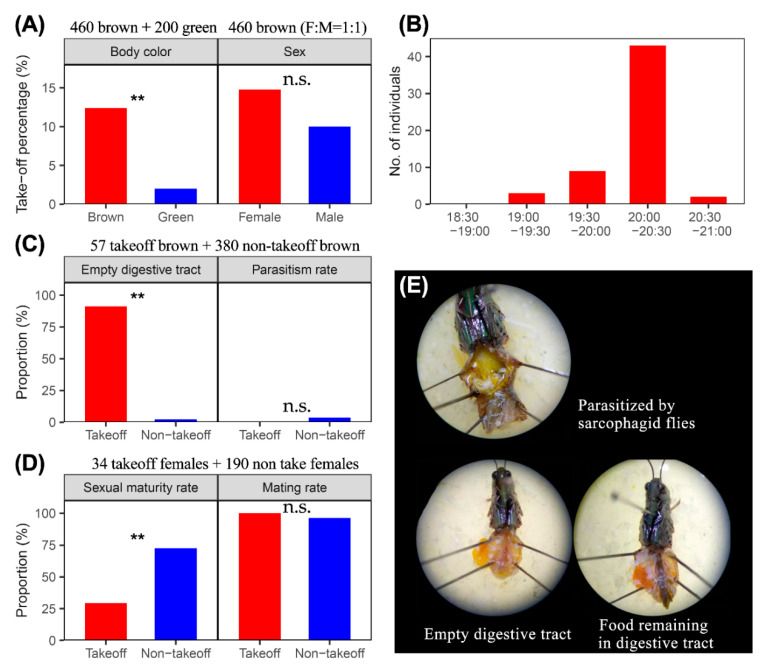
Take-off observation of *Oedaleus asiaticus* in field cages in inner Mongolia, northern China. (**A**) Take-off percentage between different body colour morphs, and between sexes; (**B**) Take-off numbers at different time periods; (**C**) Proportions of individuals with empty digestive tract or parasitized among take-off and non-take-off groups; (**D**) Proportions of individuals that had reached “sexual maturity” (Level III ovarian development or above) or mated among take-off and non-take-off groups; (**E**) Dissected adult grasshoppers showing parasitism by a sarcophagid fly, an adult with an empty digestive tract, and an adult with food remaining in the digestive tract; “**”=significant, *p* < 0.01; “n.s.” = not significant.

**Figure 2 insects-11-00416-f002:**
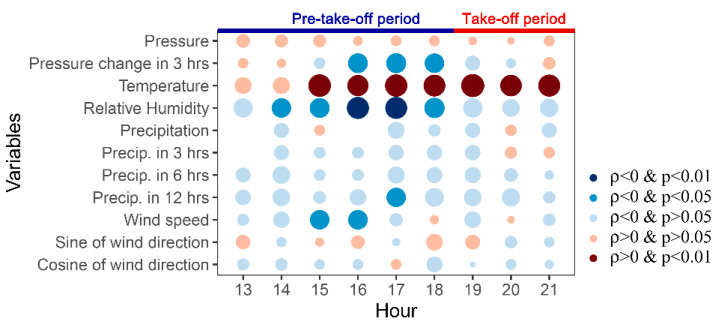
Spearman’s correlation coefficients (ρ) between the daily numbers of *O. asiaticus* taking off and several meteorological factors in the afternoon (pre-take-off) and evening.

**Table 1 insects-11-00416-t001:** Results of the generalized linear model (GLM) based on a Poisson distribution testing the effect of weather conditions on the daily number of grasshoppers taking off.

Independent Variable	Estimate	Standard Error	*z*-Value	*p*-Value
Intercept	−3.85	2.07	−1.862	0.0626
Total precipitation in last 12 h at 17:00 h	−2.49	1.55	−1.608	0.1077
Temperature at 15:00 h	0.203	0.06	3.138	0.0017
Wind speed at 15:00 h	−0.369	0.69	−5.278	<0.0001

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
