# Peer review of "Migratory Take-Off Behaviour of the Mongolian Grasshopper Oedaleus asiaticus"

_insects, 2020, doi:10.3390/insects11070416_

Round 1
Reviewer 1 Report
There are some questions regarding methodology and results, and several suggestions on the writing style. Minor suggestions on the style to write references.
|
Page |
Lines |
Observations/ Suggestions |
|
3 |
95 |
Dimensions of indoor net cages? Temperature and relative humidity of the indoor? |
|
|
97 |
How many days the experiment was carried out? |
|
|
114-115 |
Digestive tract of no food individual it does not seems a pink color, this image need more definition |
|
|
123 |
Was the air temperature calculated as mean number? |
|
5 |
152 |
Were adults or nymphs of the grasshopper caught? |
|
|
162 |
Add an “h” after 21:00 |
|
|
164 |
Add an “h” after 20:30 |
|
|
166 |
Replace: and, with: related to |
|
|
172 |
Any idea of what species is the sarcophagid fly? |
|
|
172 |
Replace: no; with: any of |
|
|
183-186 |
Spearman’s correlation coefficient was already explained in Materials and Methods (lines 119-122) |
|
|
186 |
Any idea of this range of air temperature? |
|
|
191 |
What was the range of the warmer air temperature? |
|
|
192 |
Range of the higher relative humidity? |
|
|
195 |
What you mean with “fine weather”? |
|
6 |
201 |
Add: (Figure 2), after significantly |
|
|
204 |
What do mean with “large nocturnal migratory insects”? |
|
|
213 |
How is the behavior of empty digestive tract in other migratory grasshopper species? |
|
|
239 |
Are the grasshoppers stop eating the day are going to take-off? |
|
7 |
249-250 |
What species are examples of other migratory insects? |
|
|
267 |
Add mated and, before sexually |
|
|
267 |
You may add: specially temperature, after conditions |
|
|
275 |
There are some observations on the style to write references, e.g. a space between volume and numbers of pages (line 277 vs. 285) |
Author Response
Referee: 1
P3 L97-100: Dimensions of indoor net cages? Temperature and relative humidity of the indoor?
>>> More information was added. “(a cylinder cage with diameter 24 cm and height 26 cm)”. / “To keep the small net cages close to natural state, no temperature and humidity measuring instruments were used indoors.”
P3 L101: How many days the experiment was carried out?
>>> “Our experiment was carried out during 14 July-6 August, in total 24 days.”
P3 L118-120: Digestive tract of no food individual it does not seems a pink color, this image need more definition
>>> Under the microscopes, ‘No Food’ digestive tract was pink and transparent. The pink color was very light, and thus was not obvious in the image. We identified a ‘No Food’ digestive tract mostly by seeing whether it was transparent, so we removed the word ‘pink’ to avoid misunderstanding.
P3L129: Was the air temperature calculated as mean number?
>>> In previous submission, the wind speed and direction were the value on the hour, not the mean value during one hour. Now, all variables except precipitation are the mean value in one hour, and the result of wind variables has been updated. To find out which variable is the most important for predicting the take-off grasshopper in the evening, General Linear model was further applied. The optimized model indicated the air temperature and wind speed at 15:00h were the most important variables. We think this new result will be very helpful, so this new result was also added.
P5 L170: Were adults or nymphs of the grasshopper caught?
>>> More than 660 grasshoppers (adults and nymphs) were caught from wild and fed in net cages. But only 660 adults were used in our experiments. To make this clearer and simpler, the sentence was changed to “From 14 July to 6 August 2019, 660 adult grasshoppers were used in the experiments, of which 200 were the green-coloured morph.”
P5 L181/183: Add an “h” after 21:00 & 20:30
>>> Added.
P5 L190: Replace: and, with: related to
>>> Changed.
P5 L196: Any idea of what species is the sarcophagid fly?
>>> Sorry, we have no information about the species.
P5 L196: Replace: no; with: any of
>>> “no take-off individuals were parasitized” was changed to “but none of take-off individuals”
P6 L207: Spearman’s correlation coefficient was already explained in Materials and Methods (lines 119-122)
>>> Deleted.
P6 L214: Any idea of this range of air temperature?
P6 L218: What was the range of the warmer air temperature?
P6 L220: Range of the higher relative humidity?
>>> All this information was added.
P5 L214-215: “mean temperature: 25.7±0.6°C (range 20.0-29.7°C, n=24 days)”
P6 L218-219: “period (mean: 29.3°C, range: 23.4-33.4°C, n=24 days)”
Apart from the above information added, we added a small paragraph at the beginning of this section to describe the general weather conditions during our experimental periods. We think this information will help the reader to understand.
P5 L207-211: “During 14 July-6 August 2019, the weather in Xilinhot was quite dry, the daily relative humidity was only 49.1% (range 30.1%-64.8%). The mean daily temperature was 24.7±0.4°C (mean ± stand error (S.E.), n=24 days), but the difference between day and night was large (13.2±0.5°C, n=24 days). Thus, the daily max temperature was up to 31.2±0.5°C (n=24 days). Surface wind blew towards the south (Rayleigh test; direction 187°, speed 3.3±0.2 m/s, r = 0.59, p =0.0001, n=24 days).”
P6 L224: What you mean with “fine weather”?
>>> changed ‘fine’ to ‘calm’.
P6 L230: Add: (Figure 2), after significantly
>>> Added.
P7 L245: What do mean with “large nocturnal migratory insects”?
>>> “nocturnal migratory insects with the similar or larger body size” of O. asiaticus. Some smaller nocturnal migratory insects have different behaviour. Like the brown planthopper which mostly takes off at dusk in warmer conditions, but takes off earlier in cooler weather.
P7 L252: How is the behavior of empty digestive tract in other migratory grasshopper species?
We assume this happens but we can’t find any reference to it, other than the cited Lambert (1972) paper (P7 L264). He found that, in C. terminifera, the foregut of insects taking-off after sunset was markedly less full than that of insects remaining on the ground.
P6 L277: Are the grasshoppers stop eating the day are going to take-off?
>>> We did not have chance to look at this. “During daytime, grasshoppers were observed to be very active, and can easily be simulated to take off by any disturbance, such as humans approaching the cage or other individuals flying. However, in this ‘trivial’ flight activity, individuals make horizontal flights at a low height (< 1m) and land in a short time, so this behaviour was obviously different from the migratory take-off at dusk.” (P5 L185-189).
P7 L289: What species are examples of other migratory insects?
>>> added “(e.g. Mythimna separata [36] and Nilaparvata lugens [37])”
P8 L307 and 308: You may add: specially temperature, after conditions
>>> “especially temperature and wind speed” was added
P8 L307: Add mated and, before sexually
>>> Added
P8 L318: There are some observations on the style to write references, e.g. a space between volume and numbers of pages (line 277 vs. 285).
>>> Checked again, and corrected.
Reviewer 2 Report
Review of the ms “Migratory take-off behaviour of the Mongolian grasshopper Oedaleus asiaticus”
This is an interesting paper, well written, worth publishing. I have just a few comments/suggestions indicated here below.
L. 41-43: Just a few comments. It is now well known that locusts (both in the solitary and gregarious phase), but also many grasshopper species, undertake regular seasonal migrations according to weather conditions, take-off taking place at dusk for grasshoppers and solitary locusts (the gregarious locusts migrating during the day). The understanding of such a bevahiour is fundamental to built forecasting systems to try to prevent outbreaks and damage.
It would be nice to add the two following references:
- For a general overview of flight and migration in Acridoids:
Farrow R.A., 1990. Flight and migration in Acridoids. pp. 227-314. In: Chapman R.F. & Joern A. (Eds) The Biology of Grasshoppers. John Wiley, London.
- For two examples of forecasting systems in the Migratory locust (Locusta migratoria) and the Senegalese grasshopper (Oedaleus senegalensis) based on a good understanding of the migratory behaviour of these species :
Lecoq M., 1995. Forecasting systems for migrant pests. III. Locusts and grasshoppers in West Africa and Madagascar. p. 377-395. In: Drake V.A. & Gatehouse A.G. (Eds) Insect migration : physical factors and physiological mechanisms. Cambridge University Press, Cambridge.
Another point is that explaining that take-off for grasshoppers take place at dusk will be usefull to understand better the protocol explained in the paragraph “Take-off behaviour observations".
L. 82-115: Are you sure that the presence of the observer in a so small cage doesn't perturb the natural behaviour of the grasshoppers ? Would be nice to comment.
For information, I join a paper where study of the locomotor activity of the Migratory locust using an actograph under semi-natural conditions (as near as possible of the field, and with simultaneous wheather records) made it possible to identify the main phases of the locust activity during the day and to define the favorable timing and conditions for these species to take-off at dusk (photoperiod, temperature, relative humidity, age of individuals, maturity, diet) (sorry the paper is in French but the illustrations understand by themselves):
Launois M., Le Berre J.R., Lecoq M., 1976. Etude expérimentale de l'activité locomotrice du Criquet migrateur malgache dans la nature (Experimental study of the locomotor activity of the Malagasy migratory locust in the field). Annls Soc. ent. Fr. (N.S.), 12(3) : 433-451.
L. 117-188 : It would have been better to try to record at least some meteorological parameters (precipitations, air temperature and humidity…) near the field cages and not only at a meteorological station 5km away.
L. 139: I don’t understand why it is said “Take-off observation of O. asiaticus in field nets” as observations are supposed to have been conducted in “field cages” (L. 101).
L. 157-158: It is said that “the green morph of O. asiaticus was regarded as resident and were excluded from the rest of the analyses.” But it can also be assumed that the take-off conditions-weather conditions in particular-for the green and brown forms are not the same. The green form can only be considered "resident" under the specific conditions of this particular study. It would be good to raise and comment on this point. But I saw this is the case in the Discussion §.
L. 190-191 : Interesting to show thar the decision to emigrate must be made earlier in the day, depending weather conditions in the afternoon.
L. 196-201: It would be interesting to conduct such an experiment during a rainier period since we know that in many species the rains (and therefore the quality of the environment for the locust/grasshopper) have a major influence on the flight activity and the decision to take off or not for a migratory flight at dusk.
L. 217-219: In Oedeleus senegalensis the 3 generations undertake important movements: the G1 and G2 mainly toward the North in the Sahel, following the ITCZ, and the G3 southward at the end of the rainy season (Lecoq 1995).
Author Response
Referee: 2
- 41-43: Just a few comments. It is now well known that locusts (both in the solitary and gregarious phase), but also many grasshopper species, undertake regular seasonal migrations according to weather conditions, take-off taking place at dusk for grasshoppers and solitary locusts (the gregarious locusts migrating during the day). The understanding of such a behaviour is fundamental to build forecasting systems to try to prevent outbreaks and damage.
It would be nice to add the two following references:
>>> Added.
Another point is that explaining that take-off for grasshoppers take place at dusk will be useful to understand better the protocol explained in the paragraph “Take-off behaviour observations".
>>> OK. In the Introduction we have added a sentence as follows:
“Moreover, O. asiaticus was observed to take off in the evening and land in the early morning [17], indicating it makes nocturnal migrations over several nights. It is well known that many migratory grasshopper species (and solitarious locusts) commence migratory flight at dusk, and fly for a variable period during the night [20-22]. Nevertheless, compared with …” (P2 L72-74)
- 82-115: Are you sure that the presence of the observer in a so small cage doesn't perturb the natural behaviour of the grasshoppers? Would be nice to comment.
>>> “During daytime, grasshoppers were observed to be very active, and can easily be simulated to take off by any disturbance, such as humans approaching the cage or other individuals flying. However, in this ‘trivial’ flight activity, individual make horizontal flights at a low height (< 1m) and land in a short time, so this behaviour was obviously different from the migratory take-off at dusk.” (P5 L185-189). In the evening, grasshoppers are calm and not easy to be perturbed. The take-off individuals always take off straight up and can reach the height > 1m.
- 117-188: It would have been better to try to record at least some meteorological parameters (precipitations, air temperature and humidity…) near the field cages and not only at a meteorological station 5km away.
>>> We agree. But, unfortunately, we did not have records because the meteorological instrument near the field cage was broken. So alternatively, we used the data from the meteorological station 5 km away.
- 147: I don’t understand why it is said “Take-off observation of O. asiaticus in field nets” as observations are supposed to have been conducted in “field cages” (L. 101).
>>> Experiment was conducted in field cages. Thanks, we have corrected this typo.
- 157-158: It is said that “the green morph of O. asiaticus was regarded as resident and were excluded from the rest of the analyses.” But it can also be assumed that the take-off conditions-weather conditions in particular-for the green and brown forms are not the same. The green form can only be considered "resident" under the specific conditions of this particular study. It would be good to raise and comment on this point. But I saw this is the case in the Discussion §.
>>> Yes, we agree. The text now reads: “Therefore, the green morph of O. asiaticus was regarded as resident in this study and it was excluded from the rest of the analyses (although we cannot rule out the possibility that its take-off conditions were merely different from the brown morph).” (P5 L166-167)
- 190-191 : Interesting to show that the decision to emigrate must be made earlier in the day, depending weather conditions in the afternoon.
>>> Thanks.
- 196-201: It would be interesting to conduct such an experiment during a rainier period since we know that in many species the rains (and therefore the quality of the environment for the locust/grasshopper) have a major influence on the flight activity and the decision to take off or not for a migratory flight at dusk.
>>> Yes, we agree.
- 217-219: In Oedeleus senegalensis the 3 generations undertake important movements: the G1 and G2 mainly toward the North in the Sahel, following the ITCZ, and the G3 southward at the end of the rainy season (Lecoq 1995).
>>> Thanks for this information. The text now reads:
“This is different from O. senegalensis in the West African Sahel where, in each of the three generations, the whole population emigrates and can move rapidly, sometimes as much as ~350 km in a night [7,20,22].” (P7 L257-258)